# The Influence of Antioxidants on Oxidative Stress-Induced Vascular Aging in Obesity

**DOI:** 10.3390/antiox12061295

**Published:** 2023-06-17

**Authors:** Hiva Sharebiani, Shayan Keramat, Abdolali Chavoshan, Bahar Fazeli, Agata Stanek

**Affiliations:** 1Vascular Independent Research and Education, European Foundation, 20157 Milan, Italy; hivasharebiani@yahoo.com (H.S.); shayan.sk1993@gmail.com (S.K.); ali.chavoshan73@gmail.com (A.C.); bahar.fazeli@gmail.com (B.F.); 2Support Association of Patients of Buerger’s Disease, Buerger’s Disease NGO, Mashhad 9183785195, Iran; 3Department of Internal Medicine, Angiology and Physical Medicine, Faculty of Medical Sciences in Zabrze, Medical University of Silesia, 41-902 Bytom, Poland

**Keywords:** obesity, oxidative stress, vascular aging, antioxidants

## Abstract

Obesity is a worldwide trend that is growing in incidence very fast. Adipose tissue dysfunction caused by obesity is associated with the generation of oxidative stress. Obesity-induced oxidative stress and inflammation play a key role in the pathogenesis of vascular diseases. Vascular aging is one of the main pathogenesis mechanisms. The aim of this study is to review the effect of antioxidants on vascular aging caused by oxidative stress in obesity. In order to achieve this aim, this paper is designed to review obesity-caused adipose tissue remodeling, vascular aging generated by high levels of oxidative stress, and the effects of antioxidants on obesity, redox balance, and vascular aging. It seems that vascular diseases in obese individuals are complex networks of pathological mechanisms. In order to develop a proper therapeutic tool, first, there is a need for a better understanding of interactions between obesity, oxidative stress, and aging. Based on these interactions, this review suggests different lines of strategies that include change in lifestyle to prevent and control obesity, strategies for adipose tissue remodelling, oxidant–antioxidant balance, inflammation suppression, and strategies against vascular aging. Some antioxidants support different lines of these strategies, making them appropriate for complex conditions such as oxidative stress-induced vascular diseases in obese individuals.

## 1. Introduction

Currently, obesity is considered a prevalent disease all around the world, especially in developed countries. Despite unhealthy lifestyles such as excessive fast food consumption and inactivity, some other factors, such as genetic predisposition and metabolism, can lead to obesity [1]. Obesity causes many health problems such as type 2 diabetes mellitus, cardiovascular disease (CVD), hypertension, and metabolic syndrome [2].

The level of oxidative stress has been shown to increase in obesity. It is mainly related to the production of reactive oxygen species (ROS) [3]. Therefore, many disorders that occur in obesity can be secondary to oxidative stress, and one of the most common are vascular diseases; it could be a sign of vascular aging, a prominent vascular disease [4]. In fact, vascular aging is associated with intimal thickening, increased luminal diameter, vasodilation, and endothelial dysfunction. It has been shown that this is not only age-related, and two more important mechanisms that participate in early vascular aging are inflammation and oxidative stress. Excess ROS levels have been indicated to lead to vascular aging in obesity [5].

In particular, inhibition of high levels of oxidative stress could be an effective way to prevent vascular aging in obese individuals [4,5]. In this study, our aim is to review the effect of antioxidants on vascular aging caused by oxidative stress in obesity. It seems that there is an interaction between obesity, oxidative stress, and vascular aging. To determine the best therapeutic targets, we need a comprehensive understanding of this interaction. For this aim, this review is designed to first present the mechanisms involved in obesity and oxidative stress generation. Then, vascular aging mechanisms and the role of oxidative stress in vascular cell aging are demonstrated, and eventually, different antioxidants are introduced as potential therapeutic tools related to this complex condition.

## 2. Obesity

As studies demonstrate, obesity, defined as body mass index ≥ 30 kg/m^2^, is a global crisis and requires special considerations. Recent data show that obesity incidence growth is not limited to countries of the first world, but is spreading very fast in low- and middle-income countries [6]. Obesity as a chronic process of excessive accumulation of body fat can increase the chances of chronic and metabolic diseases such as type 2 diabetes, non-alcoholic fatty liver disease, CVD, chronic kidney disease, and cancers [7,8]. Expanding adipose tissue, its remodelling, and dysfunction can induce inflammation, high levels of oxidative stress, aging, and vascular remodeling [8,9,10,11]. Therefore, obesity appears to play a significant role in vascular diseases [12,13].

Adipose tissue is a dynamic organ with different types of tissue, distribution, and function. This tissue can expand due to obesity and affect other tissues like vasculature. To better understand the dynamics of adipose tissue in obesity and its effect on the vascular system, the first section shows the physiological status of this tissue and the pathology and dysfunction of that are considered in the next section [3].

### 2.1. Types and Functions of Adipose Tissue

Cellular content of adipocytes, preadipocytes, fibroblasts, endothelial cells, macrophages, immune cells, and stromal cells: On the basis of morphology, adipose tissue can be classified into white and brown tissues. Also, in terms of location, white adipose tissue (WAT) can be subcutaneous or visceral/omental [14]. Subcutaneous WAT serves as an energy storage location, thermal insulation layer, and protective tissue against external stresses like physical stress and dermal infection [6]. Visceral fat, which people with obesity have a large amount of and can be classified into omental, mesenteric, and retroperitoneal, releases free fatty acids due to its high metabolic activity and can participate in the pathophysiology of the metabolic syndrome, hyperinsulinemia, dyslipidemia, inflammation, and atherosclerosis [6,15]. The main cell of this tissue is white adipocyte including a unilocular lipid droplet and a small number of mitochondria [13]. WAT acts as an internal secretory organ by secreting vasoactive factors and adipokines such as leptin, adiponectin, omentin, visfatin, resistin, and apelin. Hyperplasia, hypertrophy, immune cell infiltration, and vasoconstrictor secretion are changes that occur during obesity in WAT [13,16].

Brown adipose tissue (BAT) in humans contains about 4.3% of the stored fat located in the cervical, axillary, mediastinal, abdominal, and paraspinal regions [7]. It has thermogenesis, anti-inflammatory, and cardioprotective functions. Like WAT, BAT secretes cytokines such as fibroblast growth factors (FGFs), including FGF21, neuregulin 4, vascular endothelial growth factor (VEGF), and cytokines such as interleukin-6 (IL-6) [6,17]. Unlike WAT, adipocytes in BAT have multiple lipid droplets and a large number of mitochondria with an expression of uncoupling protein 1 (UCP1) in their inner membrane, which turns this tissue into a heat generator [7]. In addition to the anti-obesity role of BAT, it is resistant to induction of inflammation during obesity [7].

Perivascular adipose tissue (PVAT) is another type of adipose tissue that surrounds blood vessels. PVAT is essential for vascular homeostasis by producing adipokines and vasoactive molecules and mainly resembles the features of WAT and BAT. It has a heterogeneous phenotype depending on its location. The white phenotype is seen in the abdominal aorta and mesenteric arteries, while the brown phenotype can be seen in the thoracic aorta. Dysfunction of this tissue as a consequence of obesity can lead to generation of vascular diseases [6].

### 2.2. Adipose Tissue Remodelling

Accumulation of excessive adipose tissue leads to remodeling of this tissue. There are different aspects around adipose tissue remodelling, including adipose tissue morphology and distribution, dysregulation of cytokines and mediators in adipose tissue, infiltration of immune cells and inflammation, reduction in angiogenesis and overexpression of extracellular matrix (ECM) [13,16,18,19]. Remodeling of adipose tissue morphology is affected by hypertrophy and hyperplasia due to excessive amounts of fat [16]. Both hypertrophy and hyperplasia can extend the adipose tissue; the distribution of this expansion is one of the important parts of remodelling of the adipose tissue [19]. Accumulation of excessive adipose tissue occurs in both subcutaneous and visceral tissues, although the extension of visceral adipose tissue is greater than that of subcutaneous tissue. Excess visceral adiposity and maximum hypertrophy in adipocytes induce ectopic fat storage in the vasculature, liver, muscle, and heart [7]. In addition to distribution, the cellular remodelling of these two types of adipose tissue is different. Infiltration of inflammatory macrophages and T-CD8+ in visceral tissue is greater than that in subcutaneous tissue, whereas the number of T-helper and T-reg is decreased in visceral adipose tissue [13]. Among the immune cells infiltrated in adipose tissue, macrophages are the most important. The expansion of adipocyte tissue leads to some changes in the environment of adipocytes, including hypoxia, cell death of adipocytes, increased chemokine secretion, and dysregulation of fatty acid fluxes that requires macrophage recruitment. Therefore, macrophages play a key role in adipose tissue remodelling [19].

Dysfunction of adipose tissue due to obesity and its related consequences such as oxidized low-density lipoprotein (ox-LDL) production can cause an alteration in adipokines [3]. Dysregulation of leptin, tumor necrosis factor α (TNF-α), IL-6, IL-8, adiponectin, adipsin, resistin, visfatin, and chemerin is one of the main parts of adipose tissue remodeling. Up-regulation of pro-inflammatory adipokines, in addition to inducing an environment susceptible to inflammation, leads to vascular insulin resistance, impairment of vascular tone, vascular stiffness, and dysfunction in adipose tissue [20]. Obesity and particularly elevated levels of ROS, TNF-α, IL-6, and visfatin down-regulate the secretion of adiponectin [21]. Adiponectin and resistin are adipokines that participate in glucose metabolism [3]. Reduction in adiponectin can cause leukocyte adhesion, decreased bioavailability of nitric oxide (NO), and gluconeogenesis [11,21]. In addition, adiponectin is an anti-inflammatory mediator that acts by changing macrophages from pro-inflammatory to anti-inflammatory M2. Therefore, decreasing this mediator favors inflammation [18]. Resistin participates in appetite regulation, insulin resistance, and energy balance. Elevated levels of resistin in people with obesity are associated with a decrease in endothelial nitric synthase (eNOS) expression and, consequently, NO production mediated by oxidative stress [21]. Adipose tissue is an important organ that can produce plasminogen activator inhibitor (PAI)-1 by macrophages and adipocytes. There are different PAI-1 inducers that can elevate the levels of PAI-1 during obesity such as transforming growth factor (TGF-β), pro-inflammatory mediators like TNF-α, hypoxia-inducible factor (HIF-1α), oxidative stress and dysregulation of the circadian clock [22,23]. In addition, macrophages infiltering into adipose tissue during obesity can be the main source of PAI-1 production [23]. Leptin is a hallmark of obesity and an important appetite suppressant that acts as a signal to the brain, passes through the blood–brain barrier (BBB), and affects the hypothalamus to deliver information about fat storage. During obesity, high serum leptin levels develop resistance to leptin in the BBB transporter which leads to a decrease in leptin reaching the brain [24]. Leptin resistance as a consequence of obesity is a mechanism for decreasing sensitivity to leptin actions. Leptin can be prothrombotic and is related to CVD, especially atherosclerosis. For example, elevated leptin can cause an increase in PAI-1. Furthermore, structure-based leptin resembles pro-inflammatory cytokines and also induces the production of pro-inflammatory cytokines. Furthermore, there is an interaction between leptin and C-reactive protein (CRP) based on an adipo-hepato axis in which leptin can up-regulate CRP and CRP inhibits the functions of leptin [25].

Another important part of adipose tissue dysfunction is PVAT dysfunction caused by mechanisms like hypoxia, inflammation, and oxidative stress. Infiltration of macrophages and T cells, decreased NO production, and antioxidants are other obesity-induced alterations [9]. The most significant part of PVAT dysfunction is the loss of its anticontractile effect, which can cause pathological situations [3]. Dysregulation of PVAT as a consequence of obesity leads to dysregulated production of adipokines and cytokines such as decreased hydrogen sulphide, NO, and adiponectin and increased leptin, IL-6, TNF-α, IL-12, hydrogen peroxide, and the renin–angiotensin–aldosterone system [9,26]. The main components of adipose tissue remodelling are summarized in Figure 1.

## 3. Vascular Aging

Aging is associated with the gradual accumulation of irreversible changes that lead to cell senescence and eventually death. Senescence stops the proliferation of damaged cells, eliminates harmful factors, and prevents malignant transformation [27]. Since every organism is exposed to endogenous and exogenous damages, it is occupied with protective mechanisms including repairment mechanisms and eliminative mechanisms (autophagy, senescence, and apoptosis) [27]. An important sign of vascular aging is endothelial aging. Endothelial cells play an important role in maintaining vascular integrity and homeostasis, and they act as a barrier between blood and vascular walls. Therefore, the aging of these cells and consequently endothelial dysfunction can lead to pathologic conditions such as CVD and atherosclerosis [28]. As a result, the investigation of endothelial aging causes and mechanisms is the key concept that is perused in this section.

There are two main questions about the aging of organisms: Why do organisms age? How do organisms age? There are two perspectives to answer the first question: evolutionary and causal theory [27]. In addition, for answering the second question, there are different theories of aging including (1) structural damage theory such as the free radical theory of aging and the mitochondrial theory of aging, and (2) programmed theories of aging like genetic or cellular senescence theory of aging [26].

Different cell types have different turnover rates. The rate of endothelial cell turnover depends on its physiological or pathological conditions. Under normal conditions, it is estimated at approximately 3 years; however, this rate increases in wound healing and angiogenesis. This pattern introduces the following question: Is endothelial senescence a consequence of chronological aging or vascular disease [26]?

Answering this question depends on the answers to the general questions about the causes and mechanisms of aging. There are four categories that demonstrate the causes of aging including oxidative stress, glycation, telomer shortening, and accumulation of toxic and non-toxic products. In addition, the main mechanisms of cellular aging include mitochondrial damage and dysfunction, oxidative stress-related damages, replicative senescence due to telomere shortening, accumulation of modified proteins due to glycation, peroxidation, racemization, deamination, alkylation reactions, and toxicity of accumulated heavy metals in the cells.

Since endothelial cells are a protective barrier between the blood and vascular layers, they are more exposed to exogenous stresses such as toxic molecules and oxidants. Also, because of the strategic condition of endothelial cells, they are part of systemic changes. It seems that for a low turnover rate cell type such as that of endothelial cells, replicative senescence should not be the main mechanism of aging. In addition, due to the low proliferation rate, these cells are more prone to oxidative stress and accumulated toxins, because they cannot dilute them as a consequence of division [29]. Therefore, it seems these are the main causes of aging in endothelial cells.

### 3.1. Oxidative Stress

Oxidative stress is one of the main factors that can cause endothelial aging. Both endogenous and exogenous sources can generate oxidative stress. The activity of immune cells, inflammation, ischemia, infection, cancer, excessive exercise, mental stress, obesity, dyslipidemia, and aging are endogenous sources. Exogenous free radical production can occur as a result of exposure to environmental pollutants, certain drugs (cyclosporine, tacrolimus, gentamycin, and bleomycin), chemical solvents, cooking (smoked meat, used oil, and fat), cigarette smoke, alcohol, and radiations [30].

There is a strategic point in the cell that seems responsible for causing oxidative stress in the cell: mitochondria. Mitochondria are not only the main source of generating free radicals, but also the main sites of the antioxidant system. Therefore, mitochondrial dysfunction is directly related to elevated levels of oxidative stress. However, there is a mutual interaction between the generation of ROS and mitochondrial dysfunction. It means that elevated levels of ROS lead to mitochondrial dysfunction which can generate more ROS [2]. Mitochondria have a limited capacity for DNA repairment systems and their DNA is not covered by histones or any membrane. Therefore, the main mechanism of ROS mitochondrial dysfunction is DNA damage. It is called the mitochondrial damaging theory of aging [29]. However, endothelial cell mitochondria are not the primary source of ROS formation because of their small ratio in the endothelial cell by mass. There are other sources including peroxisomal β-oxidation of free fatty acid, xanthine oxidase, lipoxygenase, nicotinamide adenine dinucleotide phosphate (NADPH) oxidase (NOX), microsomal P-450 enzymes, cyclooxygenases, myeloperoxidase, uncoupled eNOS and pro-oxidant heme molecules [31,32,33]. Among these sources, ROS generation due to NOX isoforms that contribute to endothelial dysfunction in CVD is an important source of ROS in the endothelial cell [34,35]. In addition, the elevation of oxidative stress due to the enhancement of NOX has been observed in the PVAT of obesity-related conditions [33]. Another significant source of oxidative stress in endothelial cells is xanthine oxidase in purine metabolism which catalyzes hypoxanthine to xanthine by oxidation. The improvement of vasodilation in hypercholesterolaemic patients in the early stages of atherosclerosis due to suppression of oxypurinol-induced xanthine oxidase activity indicates the role of this enzyme in vascular disease [35]. Last but not least is eNOS as a significant source of oxidative stress in endothelial cells which generates peroxynitrite due to the uncoupling of eNOS [31]. In addition, high levels of glucose or other reducing sugars can introduce advanced glycation end products (AGEs) into the circulation, and these AGEs can bind and stimulate endothelial cells, generating ROS [36].

#### Effects of Oxidative Stress

High levels of ROS can cause deleterious effects on molecules, structures, and functions of cells. Epigenetic and transcriptomic changes, DNA damage, and acceleration of telomer shortening, inflammation, mitochondrial dysfunction, lipid peroxidation, and protein damage are some important consequences of high concentrations of ROS in cells [32,37].

High levels of ROS can activate cell signaling pathways like mitogen-activated protein kinase (MAPK) phosphoinositide 3-kinases (PI3K) and nuclear factor-kappa B (NF-κB). Activation of these pathways leads to inflammation, up-regulation of adhesion factors, invasion of monocytes into the vessel wall, proliferation, and acceleration of aging [32,37,38]. These signals induce redox-sensitive vascular smooth muscle cell (VSMC) proliferation, hypertrophy, and apoptosis, leading to thickening and remodeling of the vascular wall [32].

In addition, a high level of ROS increases the activity of matrix metalloproteinases, which leads to polymerization of hyaluronan and degradation of proteoglycans and collagen [32].

The balance of oxidative stress and NO actions is a key point in vascular function. An unbalanced status including high levels of oxidative stress and low levels of NO lead to endothelial dysfunction. Vasoconstriction, platelet aggregation, inflammation, and matrix production are the main mechanisms of endothelial dysfunction [35]. In the presence of high levels of oxygen-free radicals, the reaction of NO with these molecules leads to a decrease in NO bioavailability and the formation of peroxynitrite. On the one hand, peroxynitrite causes the switch of NO synthase by oxidation of tetrahydrobiopterin (BH4) to NO synthase uncoupling. On the other hand, it can inactivate mitochondrial manganese superoxide dismutase (MnSOD) and consequently inhibit the detoxification of superoxide to hydrogen peroxide [39]. All of these mechanisms lead to unbalanced levels of NO and oxidative stress.

Furthermore, lipid peroxidation due to high levels of ROS leads to the production of secondary lipid peroxidation products like malondialdehyde (MDA), propanal, hexanal, and 4-hydroxynonenal (4-HNE) which can induce molecular damages. MDA appears to be the most mutagenic product of lipid peroxidation, while 4-HNE is the most toxic [40].

Eventually, maybe the most significant effect of oxidative stress could be accelerated aging. Oxidative stress can accelerate the aging process through different mechanisms including molecular damage, mitochondrial dysfunction, inducing a pro-inflammatory environment, and shortening of the telomere. Oxidative stress can cause the exportation of telomerase reverse transcriptase (TERT) from the nucleus of endothelial, which leads to a lack of its activity and endothelial aging due to telomere shortening [39]. Figure 2 provides a summary of the generation and consequences of elevated levels of oxidative stress.

### 3.2. Mechanisms of Aging and Senescence

When some factors such as oxidative stress, molecular damage, telomere shortening, and inflammation stress the cell, senescence is the cellular response. Since these stresses are prolonged and the mechanism is gradual, not targeted, and not programmed, this senescence is chronic and occurs due to cell aging [27]. The aging mechanism is a complex process that includes an imbalance in different systems such as oxidant–antioxidant, inflammation, coagulation, vasodilation, autophagy, matrix metalloproteinases (MMPs) and their inhibitors, immune system, and glucose/lipid dynamics [27,34,36,41].

Oxidative stress as the main part of the aging mechanism leads to a reduction in NO bioavailability and endothelial-dependent dilation. As mentioned later, the fundamental cellular mechanisms of these alterations are mainly mitochondrial dysfunction, increased enzyme activity involved in ROS production such as NOX, xanthin oxidase, eNOS, and shortening of the telomeric [39,42].

Chronic inflammation in the aging mechanism is associated with arterial stiffness and endothelial dysfunction [42]. Infiltration of macrophages and lymphocytes and the existence of fibrous and necrosis are the main characteristics of affected tissues [27]. The molecular basis of this process is the activation of the NF-κB pathway, the expression and presence of circulating pro-inflammatory cytokines, and adhesion molecules such as CRP, IL-6, VCAM-1, and TNF-α [42]. Pro-inflammatory conditions lead to the generation of more ROS due to the upregulation of redox-sensitive genes, such as NOX [42,43,44].

Like oxidative stress, inflammation leads to a decrease in NO bioavailability and endothelial-dependent dilation. In addition, the NF-κB signaling pathway leads to the expression of MMP-9, which is related to arterial stiffness. Alteration of TGF-β and MMPs due to a decrease in NO bioavailability and also high levels of oxidative stress and inflammation are the main parts of the stiffness mechanism [42].

An important mechanism for protecting tissues from aging due to oxidative stress and inflammation is autophagy. Autophagy removes damaged organelles such as mitochondria and also damaged molecules; therefore, it is a key mechanism in stressed cells. Although the investigation of the role of autophagy in the aging process is very complex and paradoxical, it is often shown that a decrease in autophagy can promote pathological aging [41,42,43,44,45].

### 3.3. Consequences of Aging

The most important characteristics of senescent endothelial cells are their enlarged and flattened morphology, increased polyploidy, permeability, senescence-associated-β-galactosidase (SA-β-Gal) activity, and telomere shortening [27]. Furthermore, elevated levels of p16, p21, phosphorylation of p38, fibronectin, intercellular adhesion molecule 1 (ICAM-1) and inducible nitric oxide synthase (iNOS), and also decreased NO bioavailability and glycolysis are other characteristics of senescent endothelial cells [34].

The aging of endothelial cells has a wide effect on the structure and function of vasculature. Luminal enlargement, intima-media thickening, elevated endothelium permeability, and increased vascular stiffness are some of the structural changes caused by aging. On the other hand, impairment of angiogenesis and repair mechanisms, decreased vasodilation, and high sensitivity to vasoconstrictors are the main functional alterations [31,39,46]. Furthermore, a large number of transcription factors that have changed due to aging can increase the release of endothelin and VEGF from endothelial cells and cause vascular remodelling that causes these cells to undergo atherosclerosis [39]. Figure 3 demonstrates endothelial aging mechanisms and their consequences.

## 4. Obesity and Microbiota Dysbiosis Role in Inflammation and Age-Related Vascular Dysfunction

There is an interaction between obesity, microbiota, and some pathogenesis mechanisms such as inflammation and aging. There is a bidirectional relationship between obesity and microbiota dysbiosis and both of them induce inflammation in the body [47,48]. Studies demonstrate gut dysbiosis due to obesity can affect organs like the liver and adipose tissue [49]. Also, the pro-inflammatory and pro-oxidant role of microbiota dysbiosis can lead to high levels of oxidative stress and age-related vascular dysfunction like atherosclerosis [50].

Reduction in gut microbiota diversity is associated with higher levels of pro-inflammatory mediators and also non-responder individuals to weight loss interventions [48]. Regarding type, the most important feature of gut microbiota composition in obesity is the elevated level of the ratio of Firmicutes to Bacteriodetes. In addition, there are increased levels of Bacteroidales genera such as *Lactobacillus* spp., *Bifidobacterium* spp., *Bacteroides* spp., *Enterococcus* spp., and the Enterobacteriaceae species, while Clostridia, including *Clostridium leptum*, and *Enterobacter* spp. have a reduced presence in people with obesity [51]. It seems there is an association between Firmicutes abundance and obesity [48]. Elevated levels of bacteria like *Escherichia coli* as a pro-inflammatory bacteria and decreased levels of bacteria with anti-inflammatory properties such as *Fecalibacterium prausnitzii* in people with obesity indicate the role of gut microbiota dysbiosis in inducing obesity-related inflammation. In addition, the impairment of intestinal permeability and leakage of bacterial components such as lipopolysaccharide (LPS) and microbial metabolites like short-chain fatty acids (SCFA) from the intestinal lumen into the circulation can induce an immune response in the host, leading to a pro-inflammatory state [48]. Furthermore, an abundance of bacteria involved in the alteration of gut permeability and inflammation is correlated with arterial stiffness [52].

Different variables including diet, physical activity, medications, and bariatric surgery can alter microbiota composition. Effects of gut microbiota on the brain can cause weight loss and consequently inhibit mechanisms like inflammation and age-related vascular dysfunction in people with obesity. Therefore, some interventions for the prevention and treatment of obesity such as prescribing probiotics and prebiotics can be efficient for personalized management of inflammation, oxidative stress, and vascular aging due to improving gut microbiota composition [53,54].

## 5. Antioxidant Therapy

As mentioned before, there are many mechanisms in obese individuals that increase the level of oxidative stress [1]. This increase in oxidants may be the cause of obesity-related complications, which was referred to as vascular aging in this article. Therefore, the regulation of the oxidant–antioxidant balance can be a key tool for the management of obesity-induced oxidative stress that can lead to vascular aging and related diseases [55]. Treatment of obese individuals by antioxidants seems a proper step toward this aim. However, there are some challenges in selecting the best option for therapy, because there are different mechanisms like inflammation, adipose tissue remodeling, metabolic changes, mitochondrial dysfunction, and aging involved in this complex condition [2]. Therefore, to determine the best therapeutic tool, we must consider different aspects of this complex network.

The first option is quercetin, which is a secondary plant metabolite and the basic component of the human diet. Onions are a rich source of this antioxidant, but there are other sources such as grapes, cherries, apples, mangoes, citrus fruits, buckwheat, plums, tomatoes, and tea. Quercetin can be a good option because of its multifunctionality. Quercetin has all of the antioxidants, anti-inflammatory, anti-aging, and anti-obesity properties, simultaneously [48]. It acts as a potent ROS scavenger and protects macromolecules against damage by oxidative stress. It can especially inhibit lipid peroxidation, which is an important point in obese individuals. In addition, it can prevent ROS production by suppressing NOX2. Quercetin also up-regulates the expression of superoxide dismutase (SOD), catalase (CAT), and glutathione (GSH). As a chelator for Cu^2+^ and Fe^2+^, this flavonoid demonstrates one of the aspects of its antioxidant characteristics. Inhibiting neutrophil infiltration, NLRP3 inflammasomes, and NF-κB and ROS/AMPK pathways and decreasing inflammatory mediators and TNF-α-activity are the mechanisms involved in the anti-inflammatory feature of quercetin [56]. As an anti-aging agent, quercetin selectively removes aging endothelial cells. This senolytic feature along with the vasodilatory effect due to the up-regulation of eNOS activity and also anti-hypertensive and anti-atherosclerosis effects turn this antioxidant into a cardiovascular protector [57,58].

The anti-obesity feature of quercetin represents decreasing inflammation in BAT, promoting thermogenesis and browning of WAT [59,60].

Vitamins E and C have been shown to reduce inflammation-causing oxidative stress by reducing the levels of IL-6 and hs-CRP, in addition to neutralizing free radicals. The main sources of vitamin C are fruits (such as oranges, kiwi, and grapefruit) and fruiting vegetables (mainly tomato); for the main sources of vitamin E are vegetable oils (mainly sunflower and olive), non-citrus fruits, olive, and nuts [61]. Vitamin C can reduce markers of hypoxia, endoplasmic reticulum (ER) stress and inflammation, in addition to promoting the secretion of adiponectin [62]. Moreover, vitamin C plays an anti-aging role, protecting cells from oxidative stress, chromatin damage, and telomere shortening [63]. Consumption of vitamin C or vitamin E alone has been reported to lead to an improvement in endothelial function, while combined administration of them is inefficient [64]. Furthermore, an animal study in mice indicated that vitamin E supplementation reduces the levels of leptin, resistin, IL-6, TNF-a, and PAI-1, oxidative stress, and collagen deposition in visceral adipose tissue [65].

Allium sativum, as an effective substance in garlic, increases the total antioxidant capacity with improved serum levels of MDA and SOD. In addition, garlic inhibits NF-κB activation in immune cells as a main pathway for inflammatory cytokine production [66]. Also, as an anti-obesity function, sativum decreases total fat mass and reduces obesity [67].

The main sources of resveratrol with trans-3,5,4′-trihydroxystilbene formula are some plants such as grapes, apples, blueberries, and plums. Furthermore, wine and peanuts have been revealed to be rich in resveratrol [68]. Resveratrol plays a role in reducing H_2_O_2_ production [69]. In people with obesity, decreasing the level of regulatory T cells caused by consuming high-fat foods can lead to an increase in the level of oxidative stress [70]. Resveratrol has been shown to play an important role in improving T-Regs function and preventing suppression of T-Regs production [71]. Furthermore, by changing the signaling pattern, resveratrol inhibits the MAPK pathway and therefore reduces the ROS level [72]. In addition, resveratrol has been demonstrated to inhibit the PKA and Akt/PKB pathways. Therefore, it could lead to the control of inflammation and the cell apoptotic process [73].

Resveratrol has been shown to play an important role in mitochondrial function in adipose tissue and lead to energy expenditure. Hence, resveratrol acts as a WAT remodeling to BAT. This function occurs by promoting NAD-dependent deacetylases and sirtuin-1 (SIRT1) activation and subsequently the thermogenesis effects of SIRT1 by inducing BAT [74]. As an anti-obesity agent, resveratrol has an anti-lipolytic role and decreases the accumulation of glycerol in the adipose tissue. Also, resveratrol eliminates adiposity by stimulating lipolysis by suppressing white adipogenesis [75].

In particular, resveratrol increases NO production in endothelial cells by upregulating eNOS expression and inducing eNOS activity [72]. Also, resveratrol acts as an anti-atherosclerotic factor which suppresses the synthesis of endothelin-1 and stimuli-induced smooth muscle cell proliferation. In addition, arterial stiffness can be inhibited by resveratrol [76].

Unlike for many antioxidants, the main natural source of Carnitine is animal products such as the richest red meat, poultry, fish, and dairy foods. Carnitine is also synthesized endogenously from lysine and methionine, especially in kidneys, liver, and brain [77]. Propionyl-L-carnitine, as one of the most consuming synthetic antioxidant drugs that is often used by people with obesity, can induce the production of SOD [78]. It should also be noted that L-carnitine protects mitochondria and plays a major role in detoxification, control of ketogenesis and glucogenesis, stabilization of cell membranes, and improvement of fatty acid oxidation processes and ATP production [79]. It also reduces ROS production [80]. Therefore, it can be said that in addition to the therapeutic role, L-carnitine also has a preventive role. Furthermore, carnitine prevents H_2_O_2_-induced endothelial senescence [81]. It has been implicated that L-carnitine suppresses the NF-ĸB signaling pathway and then reduces several inflammatory mediators like TNF-α, IL-1, and IL-6 in adipose tissue and some other organs [82].

L-carnitine as an antilipolytic agent increases mitochondrial β-oxidation of fatty acids and also decreases triglyceride synthesis [79]. In total, L-carnitine decreases serum cholesterol, especially Low-Density Lipoprotein Cholesterol (LDL-C), triglycerides, and free fatty acids. L-carnitine can also be claimed to play an important role as an antiatherosclerotic agent by reducing serum cholesterol levels and inhibiting inflammatory pathways in endothelial cells and other immune cells around the vessel walls [83]. The summary of antioxidants and their sources and functions is presented in Table 1.

## 6. Possible Strategies—Are the Effects of Antioxidants or Other Strategies Greater as Prevention or Treatment?

There is an interaction between obesity, the oxidant–antioxidant balance, and aging. Based on this interaction, there should be different strategies for the management of obesity and its consequences, oxidative stress and vascular aging. Since this condition is a complex network of mechanisms, it seems a combination of these different types of strategies is required. The main question here is whether the treatment of antioxidants can be an effective and sufficient strategy in dealing with vascular aging caused by obesity-induced oxidative stress. Are the effects of antioxidants or other strategies greater as prevention or treatment?

The first line of strategies is related to prevention and controlling of obesity by changing the lifestyle, including food consumption habits and physical activity (Figure 1). The second group is strategies associated with inducing alterations in the adipose tissue of people with obesity like reducing adipose tissue size, promoting thermogenesis, and browning of WAT (Figure 1). The next series of strategies aims to induce the oxidant–antioxidant balance by decreasing ROS production or ROS scavenging (Figure 2). Since inflammation is a significant pathological mechanism in this complex, management strategies for suppressing inflammation in both adipose tissue and vascular tissue seem to be the central points of this network. In addition, the last line of strategies acts against vascular cell aging either by affecting aging mechanisms or removing aging cells (Figure 3). In this section, different lines of management strategies are discussed.

There are two points of view for looking at the etiology of obesity as a worldwide crisis: proximal and distal. The proximal cause is an imbalance between energy intake and energy expenditure, and the distal cause needs an evolutionary perspective [6]. Based on this perspective, on the one hand, there is an adaptive role for obesity due to survival in famine. The genes that are responsible for this role are called thrifty genes, which provide enough fat storage and count as a selective advantage. But on the other hand, in the modern world today, where there is no famine, these genes cause obesity [84,85]. In particular, the thrifty genes theory is not definitive and confirmed, and there are other theories such as the drifty genes hypothesis in which obesity is not adaptive and is caused by a neutral evolutionary process [85].

The next theory is maladaptation to BAT requirement, which introduces obesity as a byproduct of variation in positive selection for thermogenesis by BAT development. Based on this scenario, change in energy-to-nutrient ratio in modern food led to overconsumption of energy. Now, if individuals with lower levels of BAT cannot burn off that energy, they become obese while the others with higher levels of BAT can remain lean [84,85]. Therefore, in addition to genetics, the low quality of the modern diet can be a reason for the obesity epidemic. One of the important changes in food nutrition is antioxidants. Plants produce antioxidants to defend against ROS produced during photosynthesis; therefore, plants are good sources of antioxidants, and they are an important part of the human diet. Interestingly, the Paleolithic intake of antioxidants from a plant-based diet is many times higher than that of the human modern diet [86]. Therefore, the quality of the modern diet can not only increase the amount of food consumption and risk of obesity, but also affect levels of oxidative stress and the human antioxidant system. As a result, it may be one of the significant points of a change in lifestyle, as the first line of strategies is the improvement of modern food quality.

In addition to the low quality of modern food, other environmental changes in the modern world such as population density, social competition, food abundance, and a sedentary lifestyle led to the behavioral switch hypothesis, which can explain obesity and its associated conditions in the modern world. Based on this hypothesis, obesity is the byproduct of a socioecological adaptation that changes from a reproductive strategy called “soldier” (high quantity) to a “stronger and smarter” lifestyle strategy called “diplomat” (high quality) [85]. Therefore, in addition to altering personal lifestyle, social reforms like reducing social competition and population density are significant strategies for decreasing the prevalence of obesity and its related diseases. Figure 1 shows the proximal and distal causes of obesity and its management strategies.

In addition to prevention and control of obesity, remodelling of adipose tissue in obese individuals could be a therapeutic approach along with the modulation of inflammation and oxidative stress. Alteration in cell morphology, hypertrophy, hyperplasia, hypoxia, dysregulation of adipokines, infiltration of immune cells, inflammation, high rate of fatty acid oxidation, oxidative stress, and change in the thermogenesis of BAT are the main mechanisms involved in adipose tissue remodelling. Therefore, there could be different points to consider as therapeutic targets (Figure 1).

There are different animal model studies that indicate that antioxidants can affect both oxidative stress and adipose tissue. Unlike quercetin, which has no impact on adipose tissue size, chestnut, rice bran, and crude chalaza hydrolysates digested with protease A (CCH-A) reduce adipose tissue size. Chestnut and CCH-A can decrease serum cholesterol and increase lipolysis, respectively. Furthermore, RBEE reduces the overproduction of pro-inflammatory mediators in abdominal and epidermal visceral adipose tissue [87].

Quercetin shows beneficial effects to ameliorate metabolic syndrome, obesity, and insulin resistance [60,88,89,90]. Animal model studies demonstrate that quercetin can down-regulate proinflammatory genes and up-regulate antioxidant genes in BAT of high-fat diet-induced obese mice. Since inflammation in BAT causes the loss of its ability to regulate heat generation and the development of obesity, the anti-inflammatory effect of quercetin on BAT can improve the condition of obesity [59,91]. Furthermore, human intervention studies show that quercetin supplementation decreases triglyceride levels and is associated with lowering the risk of CVD [92]. In addition, quercetin also presents beneficial effects, especially at the tissue level, which has been shown in rats with high levels of oxidative stress and inflammation caused by asthma. In fact, it has been indicated that quercetin improves oxidative stress by decreasing MDA levels and increasing TAC, CAT, SOD, and GPX levels in serum, and decreasing inflammation with the decline in IL-6 and TNFα levels [93].

Melatonin is a pineal hormone involved in circadian rhythm regulation and lipid metabolism. Along with its antioxidant and anti-inflammatory role, melatonin can convert WAT to BAT and reduce obesity by increasing the thermogenesis of BAT in obese rats [90,94,95]. Therefore, it can be a therapeutic target for improving obesity and oxidative stress, simultaneously.

Resveratrol is another antioxidant which has anti-obesity features due to the remodeling of WAT to BAT, promoting thermogenesis by activation of SIRT1 and suppressing white adipogenesis. In addition to decreasing H_2_O_2_ production and anti-obesity effects, it can modulate inflammation and induce NO production in endothelial cells. Since these mechanisms are important in vascular dysfunction, this antioxidant can be helpful for vascular diseases. It has been reported that resveratrol can perform as an anti-atherosclerotic agent that affects arterial stiffness [76]. Also, in animal mode examination, it has been shown that resveratrol has desirable cardiovascular effects, and its consumption in Wistar albino rats results in a low rate of endothelial dysfunction [96]. L-carnitine can be the next efficient antioxidant because of its anti-inflammatory and anti-obesity effects, inducing SOD production and reducing ROS, preventing aging mechanisms of endothelial cells [78,80]. Regarding the antioxidant effect, it has been demonstrated that L-carnitine has an important role in improving oxidative stress recovery and prolonging lifespan in *C. elegans* worms. It means that L-carnitine could be used in vascular dysfunction induced by high levels of oxidative stress [97].

Vitamins C and E can also suppress both inflammation and oxidative stress. Administration of these vitamins separately can improve endothelial function [64,98]. Vitamin E can be a useful antioxidant in people with obesity due to reducing both inflammatory adipokines and oxidative stress [65,99]. It has been reported that vitamin C has some anti-aging roles by protecting the cells from damage [63,100].

The pathological effects of PVAT on vascular remodeling in people with obesity turn it into one of the potential therapeutic targets for antioxidants [101]. Animal studies show that an ethanolic extract of Mangosteen pericarp (EEMP), including xanthone, affects arterial remodelling by improving thickened PVAT and down-regulation of VCAM-1. Furthermore, polysaccharide peptides have been reported to contain D-glucan as an anti-inflammatory and antioxidant agent that can improve the H_2_O_2_ levels in PVAT by upregulating SOD and catalase in rats fed a high-fat diet. Another study on mice indicated decreased levels of oxidative stress, inflammation and also improved anti-contractile effects of PVAT due to long-term consumption of melatonin [102].

In addition to all these antioxidants that target different mechanisms for the treatment of this complex condition, the circadian clock seems a strategic point, the targeting of which can be a combinational strategy. Although reactive oxygen species are well known as sources of damage to macromolecules, they can also function as signaling molecules to regulate gene expressions. Circadian clock genes are examples of these redox-regulated genes. Oxidation was one of the first driving forces for the formation and evolution of internal clocks due to the evolution of plants and photosynthesis, enabling organisms to predict changes in oxidative stress and prepare them to coordinate survival strategies [103]. Therefore, it seems that internal timing systems have evolved for synchronization between the environment and organisms and also between different systems of an organism [97]. In addition, one of the causes of age-related diseases could be dysregulation of the circadian clock due to its role in the regulation of the redox balance [104]. Therefore, we can consider circadian clocks as a hub between the environment, obesity, oxidant–antioxidant balance, and aging.

On the one hand, environmental factors such as nutrient components, feeding patterns, and exercise affect both obesity and the regulation of circadian rhythm. For example, a high-fat diet in obese individuals can change circadian clock genes [105]. On the other hand, circadian clocks are involved in metabolic health, regulation of antioxidant levels, mitochondrial dynamics, ROS production, aging, and adipose tissue remodeling [97,105,106,107,108].

There is a mutual interaction between obesity and circadian rhythm. Both WAT and BAT have a rhythmic function due to the highly active circadian clock in the adipose tissue that is dependent on energy intake and regulates adipokine secretion [109]. Obesity can change the expression of circadian genes in omental adipose tissue [110]. An animal model shows that obesity down-regulates both the eNOS and the circadian gene expression and impairs endothelial function [111]. On the other hand, circadian disruption can lead to changes in lipid metabolism, dysregulation of adipose tissue, ROS elevation, and early vascular aging [101,102,103,104]. As a result, the circadian clock as an evolutionary product of oxidation can be considered a key point between obesity, oxidative stress, and aging. Regulation of the biological clock can act as a therapeutic approach for metabolic and vascular diseases because of its anti-obese, antioxidant, and anti-aging effects. Some antioxidants like alkaloids, polyphenols (as an effective component in curcumin), melatonin, and, most importantly, flavonoids can affect circadian rhythm. Especially nobiletin as a polymethoxy flavone has been reported to be a modulator of the circadian clock [104].

## 7. Conclusions

Taken together, it appears that early aging of endothelial cells as the main mechanism of vascular diseases can be induced by inflammation, oxidative stress, and reduced NO bioavailability as a consequence of adipose tissue remodelling in obesity. Elevated levels of oxidative stress through the activation of the NF-κB, MAPK, and PI3K signaling pathways lead to inflammation, proliferation, and vascular remodelling. In addition to inflammation, inducing molecular damage, mitochondrial dysfunction, increased levels of MMP, and telomere shortening are the consequences of elevated levels of oxidative stress involved in aging. Furthermore, there are different mechanisms involved in the decrease in NO bioavailability as a result of adipose tissue remodelling, oxidative stress, and aging. These are potential targets for the treatment of vascular diseases related to oxidative stress-induced vascular aging in obesity.

In addition to social reforms and change in lifestyle for prevention and control of the obesity epidemic as the trigger of this condition, antioxidants and change in microbiota composition could be considered as therapeutic tools. However, there is an important note about antioxidant therapy to consider: although the level of antioxidants in people with obesity decreases over time, it has been shown that it is better to measure the pro-oxidant–antioxidant balance in people. In other words, due to the fact that excessive use of antioxidants can lead to an effect called the oxidative effect of antioxidants, we should be careful in prescribing and recommending antioxidants. Furthermore, it seems that the antioxidant selected for this complex condition in addition to inducing a redox balance should possess different properties such as modulation of inflammation, remodelling of adipose tissue, improving NO bioavailability, regulation of circadian rhythm, and anti-aging features.

## Figures and Tables

**Figure 1 antioxidants-12-01295-f001:**
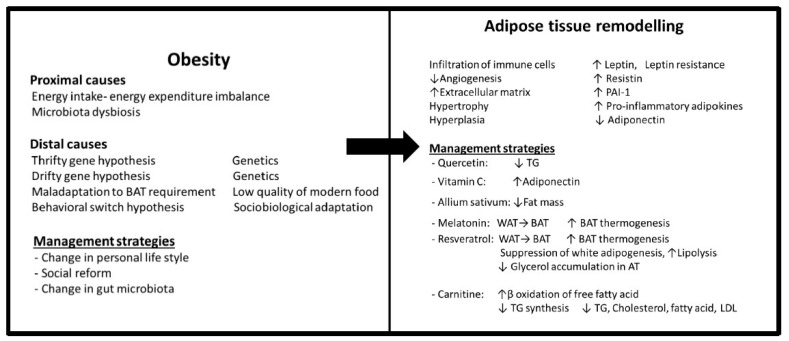
Causes of obesity, remodelling of adipose tissue and management strategies. PAI-1, plasminogen activator inhibitor-1; TG, triglyceride; WAT, white adipose tissue; BAT, brown adipose tissue; LDL, low-density lipoprotein; ↓ decrease; ↑ increase.

**Figure 2 antioxidants-12-01295-f002:**
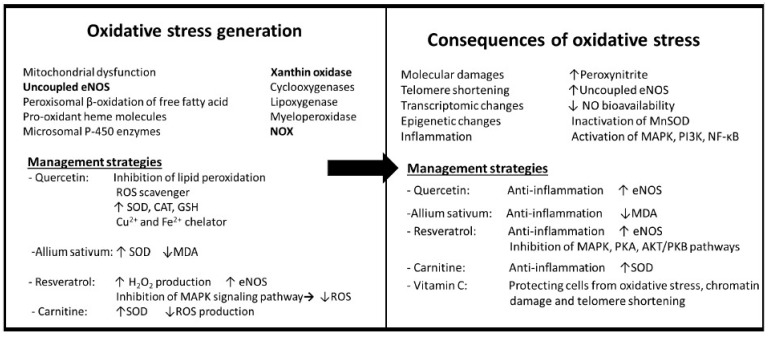
Generation and consequences of oxidative stress and management strategies. SOD, superoxide dismutase; CAT, catalase; GSH, glutathione; MDA, malondialdehyde; H_2_O_2_ hydrogen peroxide; eNOS, endothelial nitric oxide synthase; ROS, reactive oxygen species; MAPK, mitogen-activated protein kinase; PI3K, phosphoinositide 3-kinases; NF-kB, nuclear factor-kappa B; PKA, protein kinase A; PKB, protein kinase B; ↓ decrease; ↑ increase.

**Figure 3 antioxidants-12-01295-f003:**
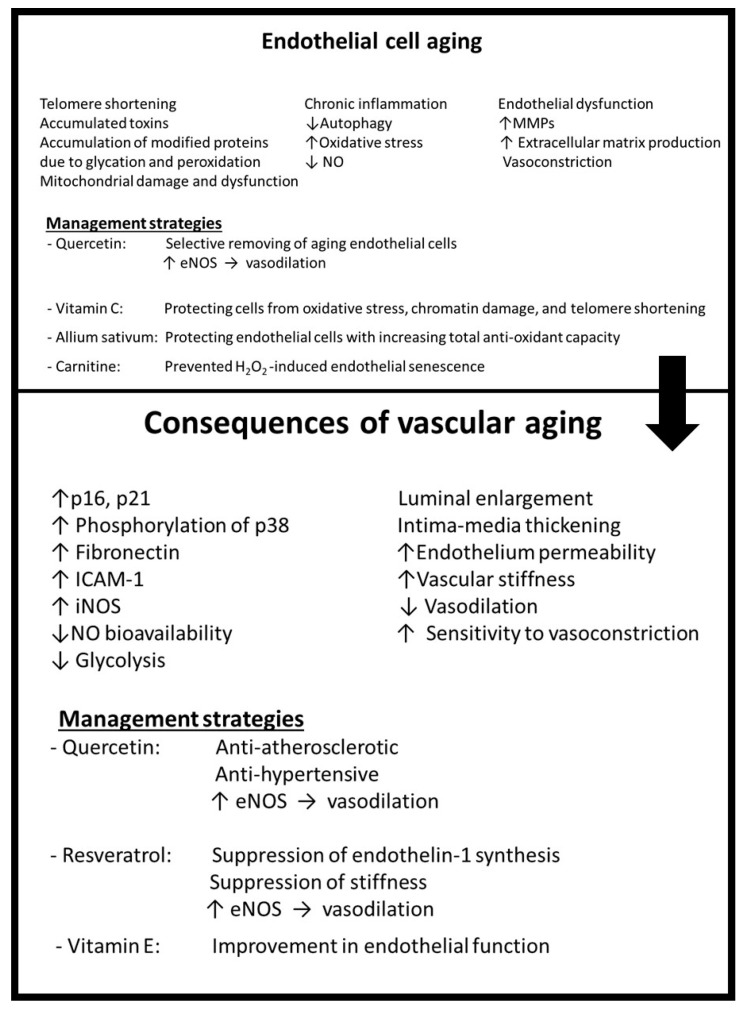
Endothelial cell aging mechanisms, vascular aging consequences and their management strategies. ICAM-1, intercellular adhesion molecule 1; iNOS, inducible nitric oxide synthase; NO, nitric oxide; eNOS, endothelial nitric oxide synthase; ↓ decrease; ↑ increase.

**Table 1 antioxidants-12-01295-t001:** Summary of antioxidants, their sources and functions.

Antioxidants	Sources	Functions			
		Antioxidants	Anti-Vascular Aging	Anti-Inflammatory	Anti-Obesity
Quercetin	Onions, grapes, cherries, apples, mangoes, citrus fruits, buckwheat, plums, tomatoes, and tea	ROS scavenger, protecting macromolecules, inhibition of lipid peroxidation, inhibiting ROS production, up-regulating the expression of SOD, CAT, and GSH	Selectively removing the aging endothelial cells, vasodilatory effect due to up-regulation of eNOS, displaying anti-hypertensive and anti-atherosclerosis effects	Inhibiting neutrophil infiltration and NLRP3, NF-κB, ROS/AMPK pathways; reducing TNF-α	Reducing inflammation in BAT, promoting thermogenesis and browning of WAT
Vitamins E and C	Vitamin C: fruits and fruiting vegetables; vitamin E: vegetable oils, olives, non-citrus fruits, and nuts	Reducing markers of hypoxia andER stress promoting the secretion of adiponectin, protecting endothelial cells from oxidative stress	Improve endothelial function, protecting endothelial cells from chromatin damages andtelomere shortening	Reducing levels of IL-6 and hs-CRP, leptin, resistin, TNF-α, and PAI-1	Reducing collagen deposition in visceral adipose tissue
Allium sativum	Garlic	Improving serum levels of MDA and SOD	Protecting endothelial cells and the endothelial function due to increasing total antioxidant capacity	Iinhibiting NF-κB activation	Reducing total fat mass
Resveratrol	Grapes, apples, blueberries, plums, wine, and peanuts	Reducing H_2_O_2_ production, increasing the level of regulatory T cells; reducing the ROS level by inhibiting the MAPK pathways	Increasing NO production, up-regulating the expression of eNOS; suppressing the synthesis of endothelin-1	Inhibiting the PKA and Akt/PBB pathway	WAT remodelling to BAT, reducing the accumulation of glycerol in adipose tissue; promoting thermogenesis by activation of SIRT1 and suppressing white adipogenesis
Carnitine	Animal products such as the richest red meat, poultry, fish, and dairy foods synthesized endogenously from lysine and methionine in kidneys, liver and brain	Inducing the production of SOD, reducing ROS production, control of ketogenesis and glucogenesis, and stabilization of cell membranes; improving fatty acid oxidation processes	Prevents H_2_O_2_-induced endothelial senescence; anti-atherosclerotic agent by reducing serum cholesterol levels	Suppressing NF-ĸB signaling pathway, reducing TNF-α, IL-1, and IL-6 in adipose tissue	Increasing mitochondrial β-oxidation of fatty acids, reducing triglyceride synthesis and serum cholesterol, LDL-C, triglycerides, and free fatty acids

ROS—reactive oxygen species; SOD—superoxide dismutase; CAT—catalase, GSH—glutathione; TNF-α-tumor necrosis factor α; IL—interleukin; hs-CRP—high-sensitivity C-Reactive Protein, ER—endoplasmic reticulum; PAI-1—plasminogen activator inhibitor-1; MDA—malondialdehyde; WAT—white adipose tissue; BAT—brown adipose tissue; H_2_O_2_—hydrogen peroxide; eNOS—endothelial nitric oxide synthase; LDL-C—low density lipoprotein, cholesterol.

## Data Availability

We used PubMed and Web of Science to screen articles for this narrative review. We did not report any data.

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
