# Peer review of "The Influence of Antioxidants on Oxidative Stress-Induced Vascular Aging in Obesity"

_antioxidants, 2023, doi:10.3390/antiox12061295_

Round 1

Reviewer 1 Report

Dear authors,

After the review process, I have several comments: figure 1 should be redesigned because it does not mean anything - the design is unproper for scientific papers poor and unclear; the paper has a significant lack of information essential for oxidative stress management - the microbiota bioactivity that is directly connected with cytokines production and obesity prevalence; the authors should include o new section related to the obesity, microbiota dysbiosis, and degenerative pathogenesis; increase in proinflammatory cytokines and inflammation, was supported by an increase in Enterobecteriaceae and the different physiological dysfunction (degenerative process) is closely linked to the pathogenesis of inflammatory diseases (e.g., obesity or IBD).

Best regards!

Reviewer 2 Report

The focus of this review is stated to be the role of oxidative stress, and more specifically antioxidants, in mediating obesity-induced vascular aging.  Overall, the manuscript is not well-organized and it is difficult to ascertain the specific hypotheses being addressed since the narrative often jumps back-and-forth from one topic to another, then back to the original topic. In addition, there are large sections devoted to causes of obesity, as well as anti-obesity interventions that are not directly relevant since they do not provide a clear link to antioxidants.

When discussing the anti-oxidant effect of various compounds many do not focus on vascular cells, especially endothelium; that is, the effects of oxidative stress on the function of cells other than endothelium are prevalent. There are also limited citations to whole animal studies. Ultimately, most of the manuscript reads as a simple list of findings from various studies and there is no “story” being told.

Finally, the table that was included has some, but not all of the anti-oxidants cited in the text. Sections 4 and 5 are largely redundant.  

Overall, the English is okay, although the choice of words is often not conventional.  For example, opening the introduction with the word "Nowadays."

Reviewer 3 Report

The work is interesting and refers to the broadening of the knowledge of antioxidants as a tool to prevent and control obesity and its associated comorbidities. It delves into the molecular mechanisms involved in these benefits and their modulation in oxidative stress.

I suggest unifying all acronyms and abbreviations throughout the text (NF-KB or NFKB; H2O2 in sub-index), as well as revising the definitions (the definition of ROS has been repeated [line 212], the acronym in cardiovascular diseases [line 223] is missing, revise the typeface of the definition of NFKB [line 290]). Finally, I suggest changing the use of "obese people" to "people with obesity" to avoid defining the patient by his or her own disease. Otherwise, I believe that the publication is prepared carefully and meets the criteria for publication.

Round 2

Reviewer 1 Report

No other comments.

Reviewer 2 Report

The authors adequately addressed my concerns. Inclusion of the figures will do much to aid with following the narrative.  I have no further comments or suggestions.